# Outcomes and Future Prospect of Japan’s National Action Plan on Antimicrobial Resistance (2016–2020)

**DOI:** 10.3390/antibiotics10111293

**Published:** 2021-10-22

**Authors:** Yoshiaki Gu, Yumiko Fujitomo, Norio Ohmagari

**Affiliations:** 1Department of Infectious Diseases, Graduate School of Medical and Dental Sciences, Tokyo Medical and Dental University, Tokyo 113-8510, Japan; 2AMR Clinical Reference Center, National Center for Global Health and Medicine Hospital, Tokyo 162-8655, Japan; yufujitomo@hosp.ncgm.go.jp (Y.F.); nohmagari@hosp.ncgm.go.jp (N.O.); 3Department of Infectious Diseases, Disease Control and Prevention Center, National Center for Global Health and Medicine Hospital, Tokyo 162-8655, Japan

**Keywords:** antimicrobial resistance (AMR), public health, action plan, public awareness, surveillance, antimicrobial stewardship, One Health approach

## Abstract

The threat from antimicrobial resistance (AMR) continues to grow. Japan’s National Action Plan on Antimicrobial Resistance, which was formulated in 2016 and targets six areas, has already had a major impact on the countermeasures implemented against AMR. Particular advances have been made in AMR-related surveillance, and we now know the situation regarding antimicrobial use and antimicrobial-resistant bacteria in the country. Educational and awareness-raising activities for medical professionals and the general public have been actively implemented and seem to have contributed to a gradual move toward an appropriate use of antimicrobials. However, there is still insufficient understanding of the issue among the general public. Determining how to use surveillance results and implementing further awareness-raising activities are crucial to address this. Tasks for the future include both raising awareness and the promotion of AMR research and development and international cooperation. The government’s next Action Plan, which will detail future countermeasures against AMR based on the outcomes of and tasks identified in the current Action Plan, has been delayed due to the COVID-19 pandemic and is urgently awaited.

## 1. Introduction

Antimicrobial agents have made it possible to treat and prevent bacterial infections, thereby saving countless lives. However, the effectiveness of these agents is declining due to increase in antimicrobial resistance (AMR). It is estimated that AMR is responsible for at least 700,000 deaths a year worldwide [1]. AMR is becoming endemic in the Asia-Pacific region, especially in developing countries, and it is therefore necessary to take proactive measures to combat the threat [2]. In Japan also, AMR has increased the disease burden [3] and led to the loss of many lives [4]. In response to global trends, it is imperative that AMR be viewed as a public health issue and that Japan contribute to both domestic and international efforts.

The World Health Organization (WHO) published the Global Action Plan on Antimicrobial Resistance [5] in 2015, at a time when measures against AMR were becoming a matter of global importance, and urged member states to implement measures against AMR. In 2016, the Japanese government published the National Action Plan on Antimicrobial Resistance (2016–2020) [6], positioning countermeasures against AMR, which had long been implemented among the measures against nosocomial infections, as a public health issue in itself. To implement the Action Plan, two specialist centers were established, the AMR Clinical Reference Center at the National Center for Global Health and Medicine (NCGM) Hospital and the Antimicrobial Resistance Research Center at the National Institute of Infectious Diseases (NIID).

Japan’s Action Plan consists of six areas, each with goals and detailed measures for achieving them (Table 1). The structure of the plan is based on the WHO’s Global Action Plan but also includes mechanics for international cooperation. The Action Plan has two key characteristics. The first is that it clearly describes the concept of the One Health approach, which refers to the integrated efforts made by multiple sectors (e.g., the veterinary and environmental sectors) to protect human health. Given the considerable amount of antimicrobials administered to livestock and pets as well as the spread of antimicrobials and antimicrobial-resistant organisms in the environment, measures against AMR are not adequate if implemented in the human medical sector alone [7]. Accordingly, in the One Health approach, a variety of sectors, including the human medical sector, collaboratively pursue goals.

The second major characteristic is that the Action Plan sets target figures (outcome indices) (Table 2). For example, in the human medical sector, targets were set for resistance rates of major bacterial isolates and for antimicrobial use in clinical settings. However, these outcome indices are just a small part of the Action Plan, described on only 2 of 71 pages in the document. The Action Plan must be evaluated not only based on outcome indicators but also on the degree to which the goals are achieved in each area.

A new Action Plan was expected to start from 2021, continuing on from the first 5-year Action Plan (2016–2020), in order to continuously implement measures against AMR. However, the formulation of the next Action Plan was delayed due to the impact of the 2019 novel coronavirus (COVID-19) pandemic, and its contents have not yet been released as of this writing (October 2021). Against this backdrop, this article reviews the outcomes of the current Action Plan narratively and discusses potential future countermeasures against AMR in Japan. To our knowledge, this is the first article that summarizes the AMR-related activities and results under the National Action Plan in Japan.

## 2. Public Awareness and Education

To effectively implement measures against AMR, it is essential to increase awareness of AMR and provide educational opportunities to both medical professionals (especially physicians who prescribe antimicrobials) and the general public. In Japan, the Cabinet Secretariat has been holding the National Awareness Conference on Measures Against Antimicrobial Resistance every November since 2016 and presents awards for significant public awareness and educational activities on measures against AMR [9]. The conference was not held in 2020 because of the COVID-19 pandemic.

The AMR Clinical Reference Center at the NCGM is responsible for education and awareness-raising activities for medical professionals [10]. It organizes seminars across Japan to increase awareness of measures against AMR [11] and works to enhance e-learning for medical professionals [12]. Since its establishment in 2017, the AMR Clinical Reference Center has held 41 seminars and workshops for healthcare professionals around Japan, which have been attended by more than 3500 participants [13]. As of March 2020, a total of 5399 people, including 2686 doctors and 1285 pharmacists, had registered for the Center’s e-learning program, which also launched in 2018. The center also provides posters and brochures about measures against AMR and a digest version of the guide for appropriate use of antimicrobials. Several types of posters and brochures for the general public are created every year and displayed at medical facilities and public health centers and can also be downloaded for general use [10]. Activities for the general public include developing and launching a website for information sharing, organizing events during the public awareness month (November) every year, and providing information for the mass media [14]. The number of press reports on AMR increased to 442 in 2017, 681 in 2018, and 1375 in 2019; although this dropped to 671 in 2020, likely due to the impacts of the COVID-19 pandemic [13]. In addition to these official activities, many professional associations and scholarly societies are engaged in activities to raise public awareness. The Action Plan lists three key messaging themes that should be spread to raise public awareness: antimicrobial stewardship, infection prevention and control, including vaccination, and the One Health approach.

A questionnaire survey on antimicrobial management for doctors working at clinics across Japan revealed that they have high levels of awareness [15] and suggested that a variety of awareness-raising activities promoted the appropriate use of antimicrobials. In contrast, the importance of measures against AMR is not yet fully appreciated by the general public [16], and their level of understanding has changed little in recent years (Table 3). Additionally, the percentage of the population having the correct knowledge about antimicrobials is lower in Japan than in European countries [16].

Health literacy is also lower in Japan than in Europe [17], which may be one of the reasons for the general public’s lack of interest in AMR issues. In Japan, the low individual cost of the national health insurance system permits easy access to a variety of medical institutions. Accordingly, patients who want antimicrobials can “doctor-shop” until they find one who will prescribe them. Health education in schools is important to improve health literacy [18], but AMR issues have not been sufficiently addressed in school education so far.

Exposure to information on antibiotics and AMR from reliable sources such as specialists or public organization websites has been shown to lead to good awareness and motivation for behavioral changes [19], indicating that medical professionals need to be actively involved in communicating knowledge to the general public.

Pharmaceutical companies have conducted campaigns to promote the proper use of antimicrobial agents among the general public, but they are relatively limited in scale and may not be very effective. The involvement of pharmaceutical companies is expected to have a great effect on physicians and other healthcare professionals.

## 3. Surveillance and Monitoring

Surveillance of both antimicrobial-resistant organisms and the use of antimicrobials is essential to implement measures against AMR. In Europe, both types of surveillance have been conducted and are used as indicators for AMR control. In Japan, drug-resistant bacteria surveillance has been conducted since before the implementation of the Action Plan, when it was already known that the proportion of drug-resistant bacteria varied according to the type of bacteria compared with other countries [20]. However, there was no surveillance of antimicrobial usage before the Action Plan. One study showed that the consumption of antimicrobials in Japan was lower than that in other countries [21], a fact that was referred to in the Action Plan. However, given that inappropriate use of antimicrobials occurs to no small extent in the clinical setting [22,23], reducing the use of antimicrobials can be achieved by promoting their appropriate use for the time being.

The Clinical Laboratory Division of the Japan Nosocomial Infections Surveillance (JANIS) run by the Ministry of Health, Labor, and Welfare is responsible for surveillance of antimicrobial-resistant bacteria, and the results of surveillance were used when setting the outcome indices for the Action Plan. According to the 2019 annual report, the number of participating medical institutions was 2075, which was 24.8% of the total number of hospitals (8372) in Japan [24]. Since the Action Plan started, evaluation of the situation at individual medical institutions and in local areas of interest has been ongoing, with data being aggregated by outpatient status, by hospital size (≥200 beds and >200 beds), and by prefecture. However, it is important to note that the data compiled by JANIS do not contain clinical information, and so test results include all bacteria present, not just the bacteria causing infectious diseases.

The National Epidemiological Surveillance of Infectious Diseases Program under the Act on the Prevention of Infectious Diseases and Medical Care for Patients with Infectious Diseases includes surveillance of infectious diseases caused by antimicrobial-resistant bacteria. Infections with four types of such bacteria are subject to notifiable disease surveillance, and infections with three types of antimicrobial-resistant bacteria are subject to sentinel surveillance. Data are collected for patients with notifiable diseases but not for carriers. NIID publishes aggregate data and evaluates the results with clinical information taken into account [25].

The AMR Clinical Reference Center conducts national surveillance of antimicrobial use by aggregating data on the amount of antimicrobials sold as well as data in the National Data Base of Health Insurance Claims and Specific Health Checkups of Japan. The AMR Clinical Reference Center publishes the results based on both sources [26].

From the viewpoint of the One Health approach, data from the veterinary and environmental sectors, in addition to data from the human medical sector, need to be evaluated. The government annually publishes the AMR One Health Report, which includes downloadable tables and figures [8]. Moreover, a summary of this report along with various AMR-related surveillance data are made available on a public website [27].

The Action Plan has led to the establishment of several surveillance systems necessary for anti-AMR measures, and the results of individual surveillance efforts, which were only handled individually in the past, can now be examined in the same format. This is the biggest improvement brought about by the Action Plan. How these results can be used for effective measures is a task to be tackled in the future.

## 4. Infection Prevention and Control

Measures against AMR have long been implemented among the measures against nosocomial infections in Japan [28]. Since the 1990s, the Ministry of Health, Labor, and Welfare has been working on raising awareness of evidence-based measures and promoting the activities of infection control teams (ICTs) and collaboration among medical institutions. In 2014, the ministry announced the establishment of organizations and regional collaborations for infection control as the basic concept for measures against AMR. Moreover, because medical institutions continue to be hotspots for the emergence and spread of antimicrobial-resistant bacteria, the prevention and control for such institutions remain crucial.

A new development under the 2016 Action Plan was the Japan Surveillance for Infection Prevention and Healthcare Epidemiology (J-SIPHE) program [29]. J-SIPHE aggregates information on anti-infection measures (e.g., occurrences of healthcare-associated infections and antimicrobial-resistant bacteria and the use of antimicrobials), so that these pieces of information are taken into account when formulating measures at individual medical institutions and promoting regional collaboration. Individual institutions can enter relevant data via a webpage (semi-automated using data for reimbursement billing and JANIS feedback data) and then view comparisons against benchmarks and changes over time. Furthermore, voluntary groups can be formed to compare the situation among member institutions in the group. J-SIPHE reduces the time and effort required for data collection and management and is expected to provide benchmarks that reflect the situation of measures against nosocomial infections in Japan in the future.

## 5. Appropriate Use of Antimicrobials

A basic strategy in countermeasures against AMR is using antimicrobials appropriately, thereby minimizing the potential to induce AMR. Many medical institutions have created multidisciplinary antimicrobial stewardship teams, and since 2018, they were able to claim remuneration for creating these teams. Among these teams’ activities, performing audits and providing feedback on the treatment of infectious diseases, introducing a preauthorization system for antimicrobial use, and developing clinical practice guidelines within individual institutions have been demonstrated to be effective [30,31]. Some Japanese medical institutions have reported the activities of their antimicrobial stewardship teams [32,33,34], and a variety of further actions will be implemented by such teams in the future.

Antimicrobial use is high in the outpatient setting in Japan [35]. More than half of antimicrobials for outpatients are prescribed unnecessarily, and some of the antimicrobials prescribed for diseases that require treatment with antimicrobials are not necessarily the first-line agents [22]. Another study also reported that antimicrobials were unnecessarily prescribed for approximately 30% of non-bacterial acute respiratory tract infections [23]. Under such circumstances, promoting the appropriate use of antimicrobials in the outpatient setting with the aim of reducing antimicrobial use is an important task in the Action Plan.

In June 2017, the Ministry of Health, Labor, and Welfare published the Manual of Antimicrobial Stewardship (first edition) [36] for doctors in the outpatient setting as part of its antimicrobial stewardship education and training program. The manual describes how to judge the need for antimicrobials and select which to use with adults and children of school age or older with acute respiratory tract infections or acute diarrhea. The manual is comprehensive, and the contents cover not only the usage of antimicrobials but also the concepts behind treating acute respiratory tract infections and the points to be highlighted in explanations to patients. A new chapter on the treatment of infants was included in the Manual of Antimicrobial Stewardship (second edition) published in December 2019 [37]. The Japanese Association for Infectious Diseases published Recommendations for Appropriate Use of Antimicrobial Agents in Respiratory Tract Infections in October 2020 to promote the appropriate use of antimicrobials for patients with underlying conditions [38].

Antimicrobial use, especially oral antimicrobial use, has been gradually decreasing since the publication of the Action Plan and showed a dramatic drop in 2020 (Figure 1) [26], suggesting that many doctors are reviewing the use of antimicrobials, especially in the outpatient setting. However, the COVID-19 pandemic might have markedly influenced the results for 2020, and further monitoring is necessary. Assessing the appropriateness of antimicrobial use, not merely the quantity of antimicrobials used, is an important task in promoting their appropriate use.

## 6. Research and Development

The Action Plan includes the promotion of basic and clinical research related to AMR control as well as research to clarify the socioeconomic impact of the AMR-related diseases. The Japanese government has allocated more funds to projects and research related to AMR. In the medical field, the Ministry of Health, Labor, and Welfare has established the Antimicrobial Resistance Research Center and the AMR Clinical Reference Center as well as a surveillance system to provide a platform for promoting research. The impact of AMR in Japan has become clear as a result of these efforts [3,4]. The Ministry of Agriculture, Forestry, and Fisheries and the Ministry of the Environment have also been promoting research related to AMR, and the AMR One Health Report [8] is a summary of these achievements. The Japanese government is also working with various organizations to discuss to promote international collaborative research.

New antimicrobials and therapeutic agents with novel concepts need to be developed to tackle AMR. Although the governments of various countries and a number of international organizations have funded such research and development, the profits achieved to date do not yet cover the development costs. As a result, development remains inadequate to control the spread of AMR. A system that ensures a sufficient return on investment in drug development is needed [39]. The development of new antimicrobials has advanced, but more needs to be done [40]. Revolutionary business models should be promoted to spur development.

Another important issue is the stable supply of the antimicrobials currently used in clinical settings. A shortage of cefazolin (a first-generation cephalosporin) in 2019 caused increased demand for other antimicrobials and consequent instability in antimicrobial supply [41,42]. Many antimicrobials are generic drugs, and the production of their active pharmaceutical ingredients relies on manufacturers overseas. The Ministry of Health, Labor, and Welfare has organized meetings to make a list of key pharmaceuticals which includes antimicrobials and to put measures in place for ensuring their stable supply in Japan [43]. The government supports pharmaceutical companies that produce the raw materials and ingredients of beta-lactam antibiotics [44], and such measures are expected to lead to a stable supply of antimicrobials.

## 7. International Cooperation

Needless to say, action on AMR is an important international issue. Cross-border movement of individuals is currently limited due to the COVID-19 pandemic, but the spread of AMR across borders will attract attention again after the pandemic. Sharing information with international organizations and other developed countries and promoting measures against AMR in developing countries will help with the implementation of measures against AMR within Japan.

The Ministry of Health, Labor, and Welfare has organized the Tokyo AMR Health Conference four times since 2016. The number of participating countries was 12 in 2016, 10 in 2017, 17 in 2018, and 20 in 2021 (online). Each conference reaffirms the policy that Japan, in conjunction with the WHO Western Pacific Office, will promote measures against AMR in the Asia-Pacific region and proactively support each country’s actions [45,46,47]. At the 2016 G7 Ise-Shima Summit, which was chaired by Japan, G7 leaders set out to strengthen AMR measures. The Japanese government has been actively participating in discussions on AMR at meetings such as those of health ministers at the G7 and G20. In addition, Japan has started making the JANIS system available to other Asian countries in order to strengthen surveillance in those countries [20].

These efforts are expected to lead a range of actions in future, although the outcomes might not be seen for a while.

## 8. Have the Outcome Indices of the Action Plan Been Met?

The achievement in terms of the outcome indices is summarized in Table 4. There have been decreases in the insensitivity rate to penicillin in *Streptococcus pneumoniae* isolates and in the resistance rates to methicillin in *Staphylococcus aureus* isolates and carbapenems in *Pseudomonas aeruginosa* isolates. However, these are all still much higher than the corresponding outcome indices. The resistance rates to carbapenems in *Escherichia coli* and *Klebsiella pneumoniae* isolates were unchanged or slightly decreased, whereas the resistance rate to fluoroquinolones in *E. coli* isolates clearly increased. The development of antimicrobial resistance in Enterobacterales is of concern worldwide, and Japan is no exception. Even after the first 5-year Action Plan ended, the development of AMR had not abated. Thus, a sense of crisis may be needed in order to tackle this problem.

Antimicrobial use has decreased gradually since the Action Plan was published, with a marked drop in 2020 (Figure 1). Because of this, the uses of oral cephalosporins, fluoroquinolones, and macrolides decreased to the levels very close to the outcome indices (Table 4). According to an interrupted time series analysis of data up to 2019, efforts to implement the Action Plan were significantly associated with a decrease in total antimicrobial use [48]. In 2020, antimicrobial use decreased more than expected. The decreases in hospital visits and changes in respiratory infection trends due to the COVID-19 pandemic might have influenced this. A similar trend has been reported in the United States [49]. Therefore, it is possible that we might see a post-COVID-19 rebound. Appropriate use of antimicrobials must be promoted and monitored continuously.

As described above, at present, most of the outcome indices have not yet been met. However, the gaps are closing for many of the indices, indicating that the implementation of the Action Plan is producing some results, albeit not fully satisfactory.

## 9. Future of the Action Plan and Measures against AMR

Measures against AMR, especially the surveillance system that provides the basis of actions, have shown considerable advancements since the implementation of the Action Plan started. Although this article focuses on the human medical sector, various measures have also been taken in the veterinary and environmental sectors. Activities implemented in accordance with the Action Plan and achievements relative to the outcome indices need to be reviewed when considering future activities.

The Japanese government plans to implement further measures against AMR in the next Action Plan after 2021. However, because of the COVID-19 pandemic, formulation of the next Action Plan has not yet been completed as of this writing (October 2021). Early formulation of the next Action Plan would help with the continuous implementation of measures against AMR.

For the next Action Plan, outcome indices must be decided carefully. For example, simply reducing the total use of antimicrobials should not be the target. Any reduction would be pointless if the necessary use of antimicrobials was decreased as well. Thus, specific indices for the inappropriate use of antimicrobials need to be set based on reports from Japanese institutions. Moreover, while the decreases seen in the resistance rates to antimicrobials in bacterial isolates are favorable, the impact of these decreases will be lost if the number of cases of infectious diseases caused by these bacteria increases. Thus, more specific indices, such as the number of cases of bloodstream infections caused by antimicrobial resistance bacteria, will be more significant for clinical practice and public health. More than 8000 individuals die every year due to bloodstream infections by methicillin-resistant *Staphylococcus aureus* (MRSA) and fluoroquinolone-resistant *E. coli* [4]. The burden of bloodstream infections, especially those caused by MRSA and antimicrobial resistant *E. coli*, is high [3]. The percentage of bloodstream infections due to MRSA and *E. coli* resistant to third-generation cephalosporins, which was added as the global indicator 3.d.2 of the Sustainable Development Goals (SDGs) [50], could be a good outcome index for the Action Plan as well.

Awareness of the appropriate use of antimicrobials has risen in recent years. The importance of measures for preventing infections in the clinical settings as well as collaboration between governments and medical institutions has been recognized anew during the COVID-19 pandemic. The existing directions will be maintained by introducing actions based on new outcome indices in the new Action Plan. Furthermore, the One Health approach will continue to be followed to further promote action in the veterinary and environmental sectors, and new issues, such as a stable antimicrobial supply, will be taken into account.

Measures against AMR are an important theme requiring constant, long-term efforts. It is worth emphasizing that, globally, implementation needs to be done within the context of the SDGs [1]. This is also true in Japan. Reinforcing measures against AMR is an important long-term challenge that will lead to improved infection prevention and improved medical safety. To move forward constructively, a review is needed of the outcomes and tasks to date and areas flagged for improvement.

This study has several limitations. First, the Action Plan is extensive, and this study did not cover every aspect of it. For example, this study did not evaluate the AMR-related efforts being made by the agricultural and environmental sectors. In addition, this study is a narrative review and may not fully reflect all of the achievements of the Action Plan to date. Finally, the movement within the government toward the next Action Plan may not have been fully taken into account.

## 10. Conclusions

A variety of measures have been implemented under the Action Plan for AMR Measures published by the Government of Japan in 2016. Novel approaches have been taken in various fields, including the promotion of the One-Health approach. In the field of surveillance in particular, significant progress has been made compared with the pre-Action Plan period, including the establishment of novel surveillance systems and the launch of a website summarizing AMR-surveillance results. Antimicrobial use has decreased and is approaching target levels, but there is a need for careful monitoring due to the impact of the COVID-19 pandemic. In addition, the target percentage of drug-resistant bacteria has not yet been reached. The next Action Plan should set targets and decide what measures to take in the future based on a careful consideration of the efforts made so far.

## Figures and Tables

**Figure 1 antibiotics-10-01293-f001:**
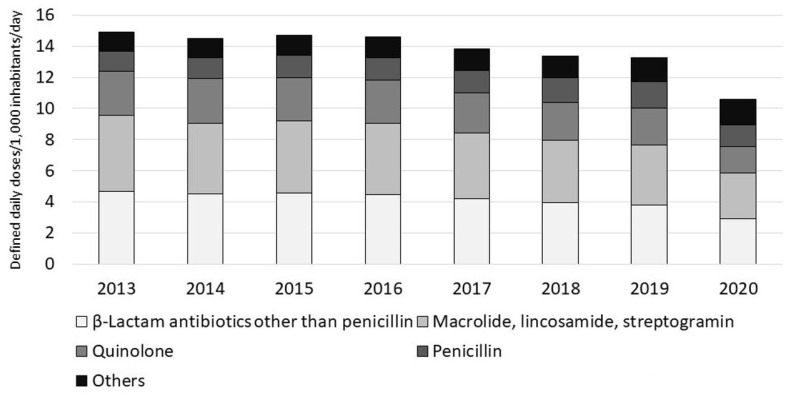
Change in antibiotics sales in Japan (2013–2020). Source: Ref. [26].

**Table 1 antibiotics-10-01293-t001:** Six areas and goals for countermeasures on AMR.

Fields	Goals
1 Public awareness and education	Improve public awareness and understanding, and promote education and training of professionals
2 Surveillance and monitoring	Continuously monitor antimicrobial resistance and use of antimicrobials, and appropriately understand the signs of change and spread of antimicrobial resistance
3 Infection prevention and control	Prevent the spread of antimicrobial-resistant organisms by implementing appropriate infection prevention and control
4 Appropriate use of antimicrobials	Promote appropriate use of antimicrobials in the fields of healthcare, livestock production, and aquaculture
5 Research and development	Promote research on antimicrobial resistance and foster research and development to secure the means to prevent, diagnose, and treat the antimicrobial-resistant infections
6 International cooperation	Enhance global multidisciplinary countermeasures against antimicrobial resistance

Source: Ref. [6].

**Table 2 antibiotics-10-01293-t002:** Outcome indices of the Action Plan (medical sector).

Resistance Rate to Antimicrobials in Bacterial Isolates		
Indices	2013	2020 (Target)
Insensitivity rate to penicillins in *Streptococcus pneumoniae* isolates	47.4%	≤15%
Resistance rate to fluoroquinolones in *Escherichia coli* isolates	35.5%	≤25%
Resistance rates to methicillins in *Staphylococcus aureus* isolates	51.1%	≤20%
Resistance rates to carbapenems in *Pseudomonas aeruginosa* isolates	17.1%	≤10%
Resistance rates to carbapenems in *E. coli* and *Klebsiella pneumoniae* isolates	0.1–0.3%	≤0.2% (similar level)
**Use of Antimicrobials (Defined Daily Doses per 1000 Inhabitants per Day)**		
**Indices**	**2013**	**2020 (Target)**
Total	14.91	≤2/3 (vs. 2013 level)
Oral cephalosporins, fluoroquinolones, and macrolides	3.91, 2.83, and 4.83, respectively	Reduction by 50% (vs. 2013 level)
Intravenous antimicrobials	0.96	Reduction by 20% (vs. 2013 level)

Prepared based on Refs. [6] and [8].

**Table 3 antibiotics-10-01293-t003:** Understanding of antimicrobials among the general public: percentage of respondents who correctly answered that antibiotics are not effective for colds and flu.

Time of Survey	No. of Respondents	Correct Answer Rate
March 2017 ^1^	3390	24.6%
February 2018 ^2^	3192	22.1%
September 2019 ^2^	3218	22.7%
September 2020 ^2^	3200	23.1%

All awareness surveys were conducted online. ^1^ Source: Ref. [16]. ^2^ Study on implementation of the National Action Plan on Antimicrobial Resistance (AMR), funded by the Health and Labor Sciences Research Grant.

**Table 4 antibiotics-10-01293-t004:** Achievement of outcome indices of the Action Plan.

Resistance Rate to Antimicrobials in Bacterial Isolates			
Indices	2013	2020	Targets
Insensitive rate to penicillin in *Streptococcus pneumoniae* isolates	47.4%	33.3%	≤15%
Resistance rate to fluoroquinolones in *Escherichia coli* isolates	35.5%	41.5%	≤25%
Resistance rates to methicillin in *Staphylococcus aureus* isolates	51.1%	47.5%	≤20%
Resistance rates to carbapenem in *Pseudomonas aeruginosa* isolates	10.7–17.1%	10.5–15.9%	≤10%
Resistance rates to carbapenems in *E. coli* and *Klebsiella pneumoniae* isolates	0.1–0.6%	0.1–0.4%	≤0.2% (similar level)
**Use of Antimicrobials (Defined Daily Doses per 1000 Inhabitants per Day)**			
**Indices**	**2013**	**2020 (vs. 2013 Level)**	**Targets**
Total	14.91	Reduction by 28.9%	≤2/3 (vs. 2013 level)
Oral cephalosporins, fluoroquinolones, and macrolides	3.91, 2.83, and 4.83, respectively	Oral cephalosporins Reduction by 42.8% Oral fluoroquinolones Reduction by 41.5% Oral macrolides Reduction by 39.5%	Reduction by 50% (vs. 2013 level)
Intravenous antimicrobials	0.96	Reduction by 2.7%	Reduction by 20% (vs. 2013 level)

Prepared based on Refs. [6,24,26,27].

## Data Availability

The datasets described in this review are available on the website of AMR Clinical Reference Center (http://amrcrc.ncgm.go.jp/index.html, accessed on 22 October 2021), Nippon AMR One Health Report (https://amr-onehealth.ncgm.go.jp/, accessed on 22 October 2021), Antimicrobial Resistance (AMR) One Health Platform system (https://amr-onehealth-platform.ncgm.go.jp, accessed on 22 October 2021), Ministry of Health, Labour and Welfare grants system (https://mhlw-grants.niph.go.jp/, accessed on 22 October 2021) and JANIS official website (https://janis.mhlw.go.jp/index.asp, accessed on 22 October 2021).

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
