# Peer review of "Outcomes and Future Prospect of Japan’s National Action Plan on Antimicrobial Resistance (2016–2020)"

_antibiotics, 2021, doi:10.3390/antibiotics10111293_

Round 1

Reviewer 1 Report

Gu et al., summarized Japan's national action plan against antimicrobial resistance. The manuscript is well structured and quite easy to read. A few minor comments that might help improve this manuscript are listed below.

1. A public awareness and education doesn't seem to be successful based on the table 3. From 2017 to 2020, correct answers don't increase. In fact 2017 correct answer rate is the highest. Authors recommend health professional and public body to advocate better but is it all due to lack of awareness? Can author discuss further from multiple perspectives? Is there anything specific to Japan's healthcare system, general public's perception or the role of pharmaceutical and health insurance company?

2. It would be interesting to see the data collected in Japan how many lives have been saved from septicemia otherwise lost from antimicrobial resistance. Extreme measures like this will provide the readers more practical understanding of action plan described in this manuscript and the manuals of antimicrobial stewardship published the ministry of health, labor and welfare in Japan. For instance, authors describes figure 1 show drastic drop but I am not sure whether that is the case. How is this significant drop?

3. I find the section 'research and development' can be further discussed. Can it be more than another 'drug development'?

Author Response

Point 1: A public awareness and education doesn't seem to be successful based on the table 3. From 2017 to 2020, correct answers don't increase. In fact 2017 correct answer rate is the highest. Authors recommend health professional and public body to advocate better but is it all due to lack of awareness? Can author discuss further from multiple perspectives? Is there anything specific to Japan's healthcare system, general public's perception or the role of pharmaceutical and health insurance company?

Response 1: Thank you for these important questions.

As you pointed out, despite the action plan, the general public’s knowledge of antimicrobials has not changed. The percentage of the population having the correct knowledge about antimicrobials is lower in Japan than in European countries. We considered the possible impacts of the health insurance system, the need for school education, and the need for pharmaceutical companies to take action.

To clarify these points, we have added the following text.

Lines 112-120

Additionally, the percentage of population having the correct knowledge about antimicrobials is lower in Japan than in European countries [16].

Health literacy is also lower in Japan than in Europe [17], which may be one of the reasons for the general public's lack of interest in AMR issues. In Japan, the low individual cost of the national health insurance system permits easy access to a variety of medical institutions. Accordingly, patients who want antimicrobials can doctor-shop until they find one who will prescribe them. Health education in schools is important to improve health literacy [18], but AMR issues have not been sufficiently addressed in school education so far.

Lines 125-128

Pharmaceutical companies have conducted campaigns to promote the proper use of antimicrobial agents among the general public, but they are relatively limited in scale and may not be very effective. The involvement of pharmaceutical companies is expected to have a great effect on physicians and other healthcare professionals.

Point 2: It would be interesting to see the data collected in Japan how many lives have been saved from septicemia otherwise lost from antimicrobial resistance. Extreme measures like this will provide the readers more practical understanding of action plan described in this manuscript and the manuals of antimicrobial stewardship published the ministry of health, labor and welfare in Japan. For instance, authors describes figure 1 show drastic drop but I am not sure whether that is the case. How is this significant drop?

Response 2:

Figure 1 is a simple tabulation of the surveillance results. To provide more information, we have cited the results of a study that performed a more detailed analysis. We also added text to the Introduction to emphasize the importance of promoting AMR measures in Japan.

The newly added text is as follows.

Lines 30-38

Antimicrobial agents have made it possible to treat and prevent bacterial infections, thereby saving countless lives. However, the effectiveness of these agents is declining due to increases in antimicrobial resistance (AMR). It is estimated that AMR is responsible for at least 700,000 deaths a year worldwide [1]. AMR is becoming endemic in the Asia-Pacific region, especially in developing countries, and it is therefore necessary to take proactive measures to combat the threat [2]. In Japan also, AMR has increased the disease burden [3] and led to the loss of many lives [4]. In response to global trends, it is imperative that AMR be viewed as a public health issue and that Japan contribute to both domestic and international efforts.

Lines 304-306

According to an interrupted time-series analysis of data up to 2019, effort to implement the Action Plan were significantly associated with a decrease in total antimicrobial use [48]. In 2020, antimicrobial use decreased more than expected.

Point 3: I find the section 'research and development' can be further discussed. Can it be more than another 'drug development'?

Response 3: Thank you for your suggestion.

In the R&D section, we focused mainly on drug discovery, but the action plan includes promoting research related to AMR countermeasures; promoting research on the diagnosis, treatment, and prevention of infectious diseases; and promoting collaboration between industry, government, and academia.

We have added the following describing the government's efforts.

Lines 240-250

The Action Plan includes the promotion of basic and clinical research related to AMR control as well as research to clarify the socioeconomic impact of the AMR-related diseases. The Japanese government has allocated more funds to projects and research related to AMR. In the medical field, the Ministry of Health, Labour and Welfare has established the Antimicrobial Resistance Research Center and the AMR Clinical Reference Center as well as a surveillance system to provide a platform for promoting research. The impact of AMR in Japan has become clear as a result of these efforts [3, 4]. The Ministry of Agriculture, Forestry and Fisheries and the Ministry of the Environment have also been promoting re-search related to AMR, and the AMR One Health Report [8] is a summary of these achievements. The Japanese government is also working with various organizations to discuss to promote international collaborative research.

Reviewer 2 Report

The article describes the outcomes and future plan of the Japan action plan on AMR  (2016-2020). The goals attained are great achievements in Japan.

The article is well-structured but hard to read.  

I consider the article highly relevant for Japan, to set future goals for the national action plan to fight AMR but it does not add to the literature globally or gives directions to other countries on how barriers and challenges are faced to implement these goals.

Author Response

Point 1: The article describes the outcomes and future plan of the Japan action plan on AMR  (2016-2020). The goals attained are great achievements in Japan.

Response 1: Thank you for your comment.

Point 2: The article is well-structured but hard to read. 

Response 2: We apologize for the inconvenience. The previous version of the manuscript was proofread by a native speaker. The other reviewers found the text clear and understandable, so we hope to proceed with the manuscript as is. However, please note that the revised manuscript has also been edited by a native speaker familiar with this area of research.

Point 3: I consider the article highly relevant for Japan, to set future goals for the national action plan to fight AMR but it does not add to the literature globally or gives directions to other countries on how barriers and challenges are faced to implement these goals.

Response 3: Thank you for your comments.

Please note that this manuscript was prepared for a special issue on AMR measures in Japan, so we considered only the Japanese approach. In the revised version, we have added descriptions to make the discussions clearer and increased the number of references. We hope that these descriptions will be useful for researchers in other countries.

Reviewer 3 Report

Nowadays, AMR is a major concern of any health system, because of its constant evolution and multiple implications. In this context, the developers of health policies makers and the responsible institutions for them have to promote a correct use of antibiotics. The aim of the present manuscript is to review the outcomes and to present the perspectives of the Japan’s National Action Plan on Antimicrobial Resistance, formulated in 2016. The authors have had access to data regarding the Japan’s National Plan and the manuscript contains a summary evaluation of the results of the AMR plan. However, the manuscript needs major improvements.

 Introduction:

  1. What sort of review is it? Is this a narrative review of literature or a systematic one? Please specify.
  2. The author should present more details on the context of the actual plan regarding AMR, including international context.
  3. The authors should compare the Japanese situation with other countries (points of similarities and differences).
  4. The authors should present the novelty of this article

Table 2

  1. Because the subject of this paper is the Plan on 2016-2020, authors should present the data for the end of 2015 not for 2013.

Public awareness and education

  1. Line 71: Authors should mention when the National Awareness Conference on Measures against Antimicrobial Resistance took place. It is not clear if this conference is organized yearly or not.
  2. Lines 76-81: The authors should present the outcomes obtained: How many seminaries or e-learning works or workshops were organized in the context of Japan’s Plan? How many participants and what type of participants were involved? How many types of posters or brochures were provided? etc.
  3. Line 85: This sentence requires a reference.
  4. Line 89: What are the measures against AMR presented in the plan? Please describe them.

Infection prevention and control

  1. Lines 155-156: Authors should present more information about this semi-automated system, including the webpage.
  2. Lines 179-190: Authors should specify if the Manual and Recommendations are included in the Plan.

Research and development

  1. What are the outcomes of the Plan in this field? In this section, the authors should present the outcomes in field 6 from the Japanese plan, not only information from literature.

International cooperation

  1. The Tokyo AMR Health Conference is the single outcome in the international cooperation field? What are the countries or institutions that participated to this conference?

Table 4

The authors should present the results at the end of 2020 and compare them with the targets and the data presented, including the end of 2015 because 2016 is the first year of the analyzed period and 2020 the last one.

Lines 243-248: The outcomes should be discussed compared to other countries.

Figure 1: you should insert the reference.

The authors should present the limitations of the present study.

The authors should include a Conclusions section for the present paper.

In the References section, there are 42 references, of which only 18 are articles. I recommend an update of this section by using articles that refer to the international situation regarding the topic of this manuscript.  

Author Response

Point 1: Nowadays, AMR is a major concern of any health system, because of its constant evolution and multiple implications. In this context, the developers of health policies makers and the responsible institutions for them have to promote a correct use of antibiotics. The aim of the present manuscript is to review the outcomes and to present the perspectives of the Japan’s National Action Plan on Antimicrobial Resistance, formulated in 2016. The authors have had access to data regarding the Japan’s National Plan and the manuscript contains a summary evaluation of the results of the AMR plan. However, the manuscript needs major improvements.

Response 1: Thank you for your comments.

Point 2: Introduction: What sort of review is it? Is this a narrative review of literature or a systematic one? Please specify.

Response 2: Thank you for your comment.

This manuscript is a narrative review. We have revised the Introduction section as follows to clearly indicate this.

Lines 74-76

Against this backdrop, this article reviews the outcomes of the current Action Plan narratively and discusses potential future countermeasures against AMR in Japan.

Point 3: Introduction: The author should present more details on the context of the actual plan regarding AMR, including international context.

Response 3: We appreciate this important advice. We have added the following paragraph accordingly.

Lines 30-38

Antimicrobial agents have made it possible to treat and prevent bacterial infections, thereby saving countless lives. However, the effectiveness of these agents is declining due to increases in antimicrobial resistance (AMR). It is estimated that AMR is responsible for at least 700,000 deaths a year worldwide [1]. AMR is becoming endemic in the Asia-Pacific region, especially in developing countries, and it is therefore necessary to take proactive measures to combat the threat [2]. In Japan also, AMR has increased the disease burden [3] and led to the loss of many lives [4]. In response to global trends, it is imperative that AMR be viewed as a public health issue and that Japan contribute to both domestic and international efforts.

Lines 50-51

The structure of the plan is based on the WHO's Global Action Plan but also includes mechanics for international cooperation.

Point 4: Introduction: The authors should compare the Japanese situation with other countries (points of similarities and differences).

Response 4: In response to your comment, we have added the following text to provide a comparison with the situation in other countries.

Lines 112-120

Additionally, the percentage of population having the correct knowledge about antimicrobials is lower in Japan than in European countries [16].

Health literacy is also lower in Japan than in Europe [17], which may be one of the reasons for the general public's lack of interest in AMR issues. In Japan, the low individual cost of the national health insurance system permits easy access to a variety of medical institutions. Accordingly, patients who want antimicrobials can doctor-shop until they find one who will prescribe them. Health education in schools is important to improve health literacy [18], but AMR issues have not been sufficiently addressed in school education so far.

Lines 137-144

In Europe, both types of surveillance have been conducted and are used as indicators for AMR control. In Japan, drug-resistant bacteria surveillance has been conducted since be-fore the implementation of the Action Plan, when it was already known that the proportion of drug-resistant bacteria varied according to the type of bacteria compared with other countries [20]. However, there was no surveillance of antimicrobial usage before the Action Plan. One study showed that the consumption of antimicrobials in Japan was lower than that in other countries [21], a fact that was referred to in the Action Plan.

Point 5: Introduction: The authors should present the novelty of this article

Response 5: We added the following sentence to describe the novelty of the article.

Lines 76-77

To our knowledge, this is the first article that summarizes the AMR-related activities and results under the National Action Plan in Japan.

Point 6: Table 2: Because the subject of this paper is the Plan on 2016-2020, authors should present the data for the end of 2015 not for 2013.

Response 6: Japan’s Action Plan sets targets by benchmarking the 2013 data that was available at the time of formulation. Because this paper describes activities based on the action plan, we have presented the data for 2013.

Point 7: Line 71: Authors should mention when the National Awareness Conference on Measures against Antimicrobial Resistance took place. It is not clear if this conference is organized yearly or not.

Response 7: We have clarified this point as follows.

Lines 83-84

In Japan, the Cabinet Secretariat has been holding the National Awareness Conference on Measures Against Antimicrobial Resistance every November since 2016 and presents awards for significant public awareness and educational activities on measures against AMR [9].

Point 8: Lines 76-81: The authors should present the outcomes obtained: How many seminaries or e-learning works or workshops were organized in the context of Japan’s Plan? How many participants and what type of participants were involved? How many types of posters or brochures were provided? etc.

Response 8: Thank you for your suggestion. We have added the following text accordingly.

Lines 89-97

Since its establishment in 2017, the AMR Clinical Reference Center, has held 41 seminars and workshops for healthcare professionals around Japan, which have been attended by more than 3,500 participants [13]. As of March 2020, a total of 5,399 people, including 2,686 doctors and 1,285 pharmacists, had registered for the Center’s e-learning program, which also launched in 2018. The center also provides posters and brochures about measures against AMR and a digest version of the guide for appropriate use of antimicrobials. Several types of posters and brochures for the general public are created every year and displayed at medical facilities and public health centers, and can also be downloaded for general use [10].

Point 9: Line 85: This sentence requires a reference.

Response 9: Because we have no concrete data from before the Action Plan, we have revised the sentence in the previous paragraph as follows.

Lines 100-102

The number of press reports on AMR increased to 442 in 2017, 681 in 2018, and 1,375 in 2019; although this dropped to 671 in 2020, likely due to the impacts of the COVID-19 pandemic [13].

Point 10: Line 89: What are the measures against AMR presented in the plan? Please describe them.

Response 10: We have added a sentence on the key messages listed in the Action Plan as follows.

Lines 103-106

The Action Plan lists three key messaging themes that should be spread to raise public awareness: antimicrobial stewardship, infection prevention and control including vaccination, and the One Health approach.

Point 11: Lines 155-156: Authors should present more information about this semi-automated system, including the webpage.

Response 11: We have added the details as follows. However, please note that the website can be accessed by only registered medical institutions, and therefore it is not appropriate to include it here.

Lines 195-198

Individual institutions can enter relevant data via a webpage (semi-automated using data for reimbursement billing and JANIS feedback data) and then view comparisons against benchmarks and changes over time.

Point 12: Lines 179-190: Authors should specify if the Manual and Recommendations are included in the Plan.

Response 12: The manual was published as part of the antimicrobial stewardship education and training program, as described in the Action Plan. We have revised the text as follows.

Lines 221-223

In June 2017, the Ministry of Health, Labour and Welfare published the Manual of Antimicrobial Stewardship (first edition) [36] for doctors in the outpatient setting as part its antimicrobial stewardship education and training program.

Point 13: Research and development: What are the outcomes of the Plan in this field? In this section, the authors should present the outcomes in field 6 from the Japanese plan, not only information from literature.

Response 13: The following paragraph was added to the R&D section based on the advice of another reviewer and also addresses this comment.

Lines 240-250

The Action Plan includes the promotion of basic and clinical research related to AMR control as well as research to clarify the socioeconomic impact of the AMR-related diseases. The Japanese government has allocated more funds to projects and research related to AMR. In the medical field, the Ministry of Health, Labour and Welfare has established the Antimicrobial Resistance Research Center and the AMR Clinical Reference Center as well as a surveillance system to provide a platform for promoting research. The impact of AMR in Japan has become clear as a result of these efforts [3, 4]. The Ministry of Agriculture, Forestry and Fisheries and the Ministry of the Environment have also been promoting re-search related to AMR, and the AMR One Health Report [8] is a summary of these achievements. The Japanese government is also working with various organizations to discuss to promote international collaborative research.

Point 14: International cooperation: The Tokyo AMR Health Conference is the single outcome in the international cooperation field? What are the countries or institutions that participated to this conference?

Response 14: We had previously mistaken the number of conferences, so we have corrected the text as well as added the number of participating countries. We have also added some text describing international cooperation.

Lines 276-278

The Ministry of Health, Labour and Welfare has organized the Tokyo AMR Health Conference four times since 2016. The number of participating countries was 12 in 2016, 10 in 2017, 17 in 2018, and 20 in 2021 (online).

Lines 281-285

At the 2016 G7 Ise-Shima Summit, which was chaired by Japan, G7 leaders set out to strengthen AMR measures. The Japanese government has been actively participating in discussions on AMR at meetings such as those of health ministers at the G7 and G20. In addition, Japan has started making the JANIS system available to other Asian countries in order to strengthen surveillance in those countries [20].

Point 15: Table 4: The authors should present the results at the end of 2020 and compare them with the targets and the data presented, including the end of 2015 because 2016 is the first year of the analyzed period and 2020 the last one.

Response 15: The figures in Table 4 have been changed from 2019 to 2020. We understand your comment regarding the use of data from 2015, but Japan’s Action Plan sets targets by benchmarking the 2013 data, so we have presented the data for 2013.

Point 16: Lines 243-248: The outcomes should be discussed compared to other countries.

Response 16: Regarding the change in antimicrobial usage, a similar trend has been reported by the US CDC. We added the following sentence accordingly.

Line 308

A similar trend has been reported in the United States [49].

Point 17: Figure 1: you should insert the reference.

Response 17: Please note that we had already inserted the reference (“Ref. 26”) in the lower right corner of the figure. However, this time we have added the appropriate reference (Ref. 6) for Table 1.

Point 18: The authors should present the limitations of the present study.

Response 18: In accordance with your comment, we added the following paragraph on limitations.

Lines 359-363

This study has several limitations. First, the Action Plan is extensive and this study did not cover every aspect of it. For example, this study did not evaluate the AMR-related efforts being made by the agricultural and environmental sectors. In addition, this study is a narrative review and may not fully reflect all of the achievements of the Action Plan to date. Finally, the movement within the government toward the next Action Plan may not have been fully taken into account.

Point 19: The authors should include a Conclusions section for the present paper.

Response 19: In accordance with your comment, we have added a Conclusion section as follows.

Lines 364-374

  1. Conclusion

A variety of measures have been implemented under the Action Plan for AMR Measures published by the Government of Japan in 2016. Novel approaches have been taken in various fields, including the promotion of the One-Health approach. In the field of surveillance in particular, significant progress has been made compared with the pre-Action Plan period, including the establishment of novel surveillance systems and the launch of a website summarizing AMR-surveillance results. Antimicrobial use has de-creased and is approaching target levels, but there is a need for careful monitoring due to the impact of the COVID-19 pandemic. In addition, the target percentage of drug-resistant bacteria has not yet been reached. The next Action Plan should set targets and decide what  measures to take in the future based on a careful consideration of the efforts made so far.

Point 20: In the References section, there are 42 references, of which only 18 are articles. I recommend an update of this section by using articles that refer to the international situation regarding the topic of this manuscript.

Response 20: In accordance with your comment, we have newly cited several articles in the revised manuscript.

Round 2

Reviewer 2 Report

Dear authors,

Congratulations on your contribution to the fight against AMR. Thank you for your reply and for addressing the comments.

Reviewer 3 Report

The Authors have addressed satisfactorily all my concerns. I have no further questions.